# Obesity and the Mediterranean Diet: A Review of Evidence of the Role and Sustainability of the Mediterranean Diet

**DOI:** 10.3390/nu11061306

**Published:** 2019-06-09

**Authors:** Santa D’Innocenzo, Carlotta Biagi, Marcello Lanari

**Affiliations:** 1Prochild Project, Department of Medical and Surgical Sciences (DIMEC), St. Orsola-Malpighi Hospital, University of Bologna, 40138 Bologna, Italy; 2Pediatric Emergency Unit, Department of Medical and Surgical Sciences (DIMEC), St. Orsola-Malpighi Hospital, University of Bologna, 40138 Bologna, Italy; carlottabiagi@yahoo.it (C.B.); marcello.lanari@unibo.it (M.L.)

**Keywords:** Mediterranean Diet, public health policy, childhood obesity, healthy lifestyle, health communication

## Abstract

Several different socio-economic factors have caused a large portion of the population to adopt unhealthy eating habits that can undermine healthcare systems, unless current trends are inverted towards more sustainable lifestyle models. Even though a dietary plan inspired by the principles of the Mediterranean Diet is associated with numerous health benefits and has been demonstrated to exert a preventive effect towards numerous pathologies, including obesity, its use is decreasing and it is now being supplanted by different nutritional models that are often generated by cultural and social changes. Directing governments’ political actions towards spreading adherence to the Mediterranean Diet’s principles as much as possible among the population could help to tackle the obesity epidemic, especially in childhood. This document intends to reiterate the importance of acting in certain age groups to stop the spread of obesity and proceeds with a critical review of the regulatory instruments used so far, bearing in mind the importance of the scientific evidence that led to the consideration of the Mediterranean Diet as not just a food model, but also as the most appropriate regime for disease prevention, a sort of complete lifestyle plan for the pursuit of healthcare sustainability.

## 1. Introduction: Mediterranean Diet, When Evidence Speaks

The Mediterranean Diet (MD) has been identified as having proved to be the most effective amongst many others in terms of prevention of obesity-related diseases [1]. In view of this scientific evidence in a prevention activity, the purpose of this review is to critically evaluate the methods that can inform and gently persuade consumers to adopt the MD principles in order to tackle childhood obesity, for the pursuit of healthcare sustainability.

MD is characterized by a high intake of vegetables, fruits, nuts, cereals, whole grains, and olive oil, as well as a moderate consumption of fish and poultry, and a low intake of sweets, red meat and dairy products [2,3]. Being poor in saturated fat intake and rich in monounsaturated fat intake, it provides a high amount of fibre, glutathione and antioxidants, and it is characterized by a balanced ratio of n-6/n-3 essential fatty acids [4,5]. The adoption of the MD on a large scale has long been reported to be protective against the occurrence of several different health outcomes [6].

Thanks to all these healthy aspects, the MD has been associated with a lower risk of cardiovascular mortality [7,8,9] and coronary diseases [10], obesity, type 2 diabetes, mellitus and metabolic syndrome in adults [11,12].

In pregnancy, a higher adherence to the MD has been associated with a lower risk of neural tube defects [13], preterm birth [14,15] and fetal growth restriction [16].

Moreover, it seems to influence the fetus’ susceptibility to gain weight later in life, as it has been associated with lower offspring waist circumference at preschool age [17] and lower offspring cardiometabolic risk [18].

How an adherence to the MD during pregnancy could have positive effects in offspring, not only during fetal life, but also later in life, is still unclear, but the induction of epigenetic modifications represents a possible explanation [19].

Despite this almost indisputable scientific evidence, some ambiguous effects highlighted in specific studies are considered depending on the definition of the MD standards and the indexes of the adhesion to the same standard used [20], so a few concerns arise when we refer to the definition of the MD, in view of the different opinions in the scientific literature [21] on how it has to be technically defined. Moreover, citizens and consumers who need to be persuaded to adopt it often have confused ideas about the foods that are part of this diet, [22]. In effect, since the MD has become an increasingly popular topic of interest worldwide, several myths and misconceptions are now associated with its nutritional pattern, and this may be a further obstacle to its correct diffusion, since the main challenges for its desiderable transferability in non Mediterranean areas largely derive from these misrepresentations.

In consideration of the scientific evidence, it seems increasingly appropriate to maintain the Mediterranean paradigm by way of an overall food composition, suggesting the possibility of implementing a Mediterranean-type dietary pattern within the context of a non-Mediterranean population. Even if the paucity of data does not give a clear picture of the health effects of the MD in non-Mediterranean countries [23], the transferability of the MD pattern to non-Mediterranean settings is desirable, and has been studied and deemed possible. However, it requires a multitude of changes in dietary habits, practical resources, and knowledge to accomplish these changes [24]. New strategies to increase the adherence to its principles among citizens are considered necessary above all due to the fact that some recent studies support the role of the MD in preventing obesity development in children [25,26,27], a thorny issue which is worth examining briefly.

Methods: This narrative review was performed with a multidisciplinary approach to identify publications about the regulatory instruments used to increase adherence to the Mediterranean Diet and stop the spread of obesity. A multidisciplinary and multilevel approach is chosen due to the nature of the topic itself: obesity is a multifactorial, multifaceted problem. Given that poor diets are a component of a more complex array of factors and owing to the complexity of the obesity epidemic, prevention strategies and policies across multiple levels and disciplines are needed in order to have a measurable effect. In order to set up population strategies prevention, combination and collaboration between different disciplines and competences has become essential to tackle this multifactorial problem. The search strategy was conducted in the following database: PubMed, Embase, and selected gray literautre sources. Reports, working papers, government documents, white papers and evaluation materials and research produced by organizations outside of the traditional commercial or academic publishing and distribution channels have been consulted for research and duly indicated in the bibliography. 

Relevant keywords relating to the Mediterranean Diet in combination with public policy terms and text words (“Public Policy”, “Mediterranean Diet,” or “diet” or “dietary pattern” “Mediterranean,” or “adherence” or “score”) were used in combination with words relating to health status (“health” or “mortality” or “morbidity,” or “cardiovascular diseases” or “obesity” and “childhood obesity”). The search strategy had no language restrictions. The date range was from the inception of the respective database until March 2019, when multiple articles for a single study were present, we used the latest publication and supplemented it, if necessary, with data from the most complete or updated publication.

## 2. Obesity as a Global Disease

According to the NCD Risk Factor Collaboration (Non Communicable Diseases Study Group’s research) [28], the rising trends in children’s and adolescents’ BMI have plateaued in many high-income countries, although at high levels, but have accelerated in some parts of Asia, with these trends no longer correlated with those of adults. Mean BMI and prevalence of obesity increased worldwide in children and adolescents from 1975 to 2016, with the rate of change in mean BMI moderately correlated with that of adults until around 2000, but only weakly correlated afterwards.

The NCD Study Group research highlights that the number of children and adolescents aged 5–19 years in the world who are moderately or severely underweight remains larger than those who are obese, showing the continued need for policies that enhance food security in low-income countries and households, especially in south Asia. The experiences of East Asia and Latin America and the Caribbean show that the transition from underweight to overweight and obesity can be rapid and can overwhelm the national capacity needed to engender a healthy transition. Furthermore, populations have changed their habits in an unhealthy nutritional shift: an increase in nutrient-poor, energy-dense foods can lead to stunted growth along with weight gain in children, adolescents, and adults, resulting in higher BMI and worse health outcomes throughout an individual’s lifespan.

In particular, obese children and adolescents are at increased risk for a large number of medical disorders, including hypertension, insulin resistance, dyslipidaemia, fatty liver disease and obstructive sleep apnoea disorder [29].

Moreover, obesity may lead not only to physical but also to psychological and social complications, such as low self-esteem, depression and stigma [30,31]. Childhood obesity is also difficult to reverse and it increases the likelihood of obesity in adulthood and the development of health problems, mainly cardiovascular diseases such as coronary heart disease, which is the leading cause of adult mortality and morbidity [32].

Therefore, it is important to prevent the onset of obesity in childhood in order to reduce the onset of obesity-related complications later in life.

With the exception of rare genetic disorders like *Prader-Willi* syndrome and some endocrine diseases like thyroid dysfunction, obesity is due to an imbalance between calorie intake and calories utilized that results in an excess of body adiposity. A sedentary lifestyle, lack of physical activity and poor dietary habits, together with genetic predisposition, all contribute to this phenomenon [33].

However, the rapid increase in BMI worldwide cannot be entirely explained by genetics; environmental factors also play an important role in childhood obesity. This is why family and school environments are so relevant to this topic. The most frequent unhealthy food choices seen in schoolchildren and adolescents include: skipping breakfast, eating at fast-food restaurants and consuming a high amount of sweets and junk food, which are calorie dense, while parents are still at work. All these behavior patterns increase the risk of childhood obesity and its comorbidities. In order to realize how much eating habits have changed in various parts of the world and how widespread the threat of obesity is, it’s worth taking a look at several recent studies on the point. In 2010, more than one third of children in the United States were classified as overweight or obese [34], but even at that time obesity and its related diseases were not limited to Western countries. In effect, mostly in the past two decades, urbanization in many Asian countries, where the prevalence of obesity was historically very low, has led to a sedentary lifestyle and overnutrition, setting the stage for the epidemic of obesity and conditions like non-alcoholic fatty liver disease [35]. In the last decade the spread of obesity has been increasing at an alarming rate, especially in China, Japan, and India [36]. In particular, China presents a unique model for weight change, as the country has experienced a shift from a history of undernutrition to a very rapid increase in obesity [37].

Some studies point out that China’s food consumption patterns and eating, along with the country’s cooking behaviors changed dramatically between 1991 and 2011: prior to the last decade there was essentially no snacking in China except for hot water or green tea, while currently the changes in cooking and eating styles include a decrease in the proportion of food steamed, baked, or boiled and a parallel increase in snacking and eating away from home. In addition to this, most recently the intake of foods high in added sugar has increased, together with a major growth in consumption of processed foods and beverages, while the profusion of supermarkets and convenience stores are dramatically changing the nature of China’s food supply, showing how dietary shifts are greatly affected by a wild Western urbanization [38,39].

The shifts in diet are also profound in the Latin America and the Caribbean region, which faces a major diet-related health problem accompanied by enormous economic and social costs [40].

Not even Africa has escaped this problem: the prevalence of overweight and obesity among children under five years of age was 5% in 2017, and in absolute numbers there has been an increase of almost 50% since 2000, from 6.6 million to 9.7 million in 2017 [41,42].

Overweight and obesity are a major public health issue in Australia, where their rates have risen over recent decades, with nearly 2 in 3 adults, and 1 in 4 children considered overweight or obese in 2014–2015. In addition, if compared with non-Indigenous Australians, Indigenous adults are more likely to be overweight or obese, and Indigenous children and adolescents are more likely to be obese. Unlike the previous examples, those who live outside of major urban areas are more likely to be overweight or obese than others [43].

It therefore appears evident that inadequate eating habits now constitute a multilevel problem threatening healthcare sustainability worldwide. In most of the aforementioned studies it is noted that consumer choices turn irremediably towards Western diets, abandoning autochthonous and typical diets. And this indisputable fact entails serious problems for healthcare sustainability [44].

## 3. The Choice: Mediterranean Diet Versus Western Diets

Insufficient levels of adherence to the MD are specifically considered to be a major factor in the concerning spread of the obesity epidemic in Southern Europe [45], where a Western type Diet that is high in saturated fats and refined carbohydrates, poor in quality and high in calorie intake is now spreading rapidly. This creates an astounding paradox, well highlighted during the European Congress on Obesity held in Vienna in May 2018, regarding the fact that within the very geographical area in which the MD was developed, there has been an extreme and undeniable shift in the population’s nutritional choices towards another diet that is full of saturated fats, sugars and processed meats [46]. According to the World Health Organization (WHO) [47], the abandonment of the MD is making children in the Mediterranean area fatter and generally less healthy than their Swedish counterparts, who from a young age become accustomed to a MD made up of more fish and vegetables.

Some recent studies report that adherence to the MD significantly began to decrease between 1961–65, whereas from 2004–2011 there was a stabilization of the trends in adherence values and even an increase among 16 countries [48]. After all, since 1961 there had been quite dramatic changes in the European Union (EU) diet: at the level of macronutrients, convergence was the most notable tendency, with Mediterranean countries increasing their intake of free sugars, saturated fats and cholesterol, while the highest-intake Northern European countries moderated their consumption of these nutrients [49].

In effect, a significant role in the Sixties was certainly played by the growing industrialization of the food market, taking into consideration that in many countries of Southern Europe these years coincided with those of the well-being generated by the consolidation of the post-war economic boom, which led to new nutritional options based on a greater quantity of processed foods and expanded availability of red meats and sugar, to the detriment of vegetables and cereals. Housewives became workers and the time to prepare meals was dramatically reduced, while ready-made meals became a common usage. The subsequent globalization and urbanization have in a few decades greatly worsened the effects of these factors [50].

Nowadays, the Westernization of the diet is particularly evident, especially among the younger generations [51]. This process, which is generally referred as the *modernization* of a society, implies a number of unhealthy lifestyle habits, not just limited to modification of food preferences toward western foods, but also relative to other common activities (electronic devices, computer and television use), which facilitated a radical shift in a population’s habits, leading to an overall imbalance between energy intake and expenditure [52,53].

In other words, the essence of the matter is that the general adoption of sedentary lifestyle and westernized dietary patterns generated the global rise of obesity and diabetes worldwide over the last few decades [54,55,56,57,58]. Large swathes of populations choose Western diets on a daily basis [59], both in Western and Southern Europe [60,61,62,63] and in developing countries [64], which is not a optimal choice for healthy living [65,66].

In effect, the rising prevalence of childhood obesity is a particularly concerning aspect of the phenomenon [67], the underlying causes of which are complex and interconnecting, involving social, cultural, familial, physiological, genetic, metabolic, and behavioral factors. The only safe and effective treatment is a real long-term change in lifestyle, as obesity is the result of unhealthy behaviors that may be difficult to change.

Table 1 reports the main dietary foods of the Mediterranean and Western Diets. The greatest difference between the Mediterranean and the Western Diets is the sources and proportion of dietary fat [2].

In view of all this, the choice between Mediterranean and Western Diets is mainly determined by a set of socio-economic, demographic and conjunctural factors. But that is not all: obesity is a complex, multifactorial, multilevel disease that not only has a significant impact on physical health but also on psychosocial well-being and therefore on quality of life. Obese people experience substantial impairments in quality of life as a consequence of their weight, and these can impact significantly on their mental health, which in turn can further impact on their physical health [68].

Clearly, important changes in the social determinants, whatever they may be, are unlikely in the absence of social, fiscal or legislative change, which remains a decisive tool for change: a brief examination of impact of such tool therefore seems appropriate.

## 4. Public Policy Measures to Tackle the Phaenomenon

Every government needs to urgently adopt policies that make a healthy diet easily accessible and affordable for all, in order to reach healthcare sustainability [69,70]. It is a fight in which time is an enemy, due to the extraordinary growth rates of the epidemic [71].

Law and regulation are now definitely considered essential tools in tackling childhood obesity [72,73,74], but unfortunately the legal interventions focused on public health are still at an early stage and the impact they may have on the obesity epidemic is not yet completely perceptible. After more than a decade of research on the topic [75,76], what is most evident is that the effectiveness of the attempt to change citizens’ behavior is subject to the quality of the public health policy choices, which should be structured in such a way as to protect citizens’ health, especially for the most disadvantaged groups. This fact calls into question the responsibility of public health policies in the planning of actions to combat obesity, from the perspective of the economic sustainability of healthcare. 

In this context, the Mediterranean-style diet is not considered a specific diet, but rather a collection of eating habits [77]. Primarily it is characterized by a substantially reduced consumption of meat, especially red and processed meat, and as not being based on a lot of sophisticated and very expensively marketed products, but centered on a very simple and humble set of foods that have to be appropriately chosen, washed and prepared with care and time, it represents a dietary plan that can be followed everywhere.

In terms of sustainability, cost-effectiveness analyses and sensitivity analyses have all shown that the MD remained highly cost effective under all scenarios [78], but the choice to adhere to its prescriptions is, however, substantially affected by the reduction of family time due to the increase in female work force participation [79]. What remains debateable today is whether the cost linked to a healthy diet that promotes the use of fish, vegetables and cereals, in place of often less expensive processed foods and ready meals, plays a significant role in citizens’ less healthy food choices [80].

Cross-sectional results [81] show how a low income is strictly associated with a poor adherence to the MD and a higher prevalence of obesity, while in the wider framework, inequalities in health are closely entwined in a worrying way with socio-economic inequality [82], especially among children. The relevance of socio-economic factors in determining such differences is evident: the structural determinants and conditions of daily life cause much of the health inequity between and within countries [83]. The creation of supportive environments for health is a basic action principle of health promotion, and equality is a connected core value.

A setting approach determines in fact an opportunity to link these two, highlighting the interplay between individual, environmental and social determinants of health [84].

This means that the environmental approach, focused on public health practice, necessarily goes through special policy actions, operating upon the intersection between Behavior, Environment, Health, and Public Policy [85].

However, current prevention programs have had little success to date and have proven ineffective in reversing the constantly rising rates of childhood obesity, so much so that the epidemic is growing rapidly even where once it did not exist, as previously discussed. If behavioral and environmental factors are so relevant, these disappointing observations reveal the urgent need to better understand the complex mechanisms involved in social conditions that lead to obesity, in order to propose better disease prevention and care.

## 5. The Three Pillars and the Public Health Choices

Although environmental factors are important, there is considerable evidence that genes also have a significant role in the pathogenesis of obesity [86], but very little can be done from this point of view with public policy. For this reason we hereby set about discussing only the interventions that can be undertaken through models such as regulation. Moreover, we must consider that there are currently three main treatment methods for obesity, which are easily identifiable in three different pillars represented by pharmacotherapy, bariatric surgery, and lifestyle modifications. The first two are treatments to be carried out on an individual basis and, due to the fact that they are extremely expensive and particularly invasive, they are commonly debated [87]. Furthermore, they are not to be taken into consideration for subjects of a developmental age, even if for some minors, bariatric surgery may be the only option to save their lives or avoid severe disease. For others, bariatric surgery may be considered morally wrong when more beneficent alternatives exist [88].

The cornerstone of lifestyle modifications includes changes to dietary and exercise habits that only public policy can effectively induce. While the first two pillars can obtain excellent results, they achieve nothing from a social point of view other than increasing social and health inequality. It is not exactly a correct use of public health policy to rely on such invasive and non-reversible surgeries, instead of intensifying a social commitment towards a substantial improvement of the conditions of disadvantaged groups [89] who are forced to live in environments that cannot prevent the onset of the disease. Moreover, the first two pillars can only help when obesity has been diagnosed, unlike the third pillar, which can also effectively be used to prevent the onset.

In this context, prevention is more necessary than ever in order to halt the increase of social and health inequality: only preventing obesity can decrease the number of years lived with diseases and tackle the spreading of the continuous rise of the epidemic. A clear policy implication is the need for careful monitoring and following-up of public health intervention, with a focus on effects—those intended or otherwise—in different socioeconomic groups [90].

One major issue is the scarcity of strong evidence on how to prevent obesity: prevention surely requires a complex, multilevel, environmental, socioeconomic, and lifespan approach, acting on changeable causal factors, such as diet or physical activity [91].

Despite a few isolated areas of improvement, no country has yet reversed its obesity epidemic [92]. However, since incontrovertibly strong scientific and experiential data from research carried out over the last two decades highlights the relevance of environmental factors in the development of obesity and how its effects can modify citizens’ lifestyle, important steps have already been made in the development of systems for the monitoring of socio-economic inequality in health. In some European member states, considerable efforts have been made over the last two decades to use existing data sources to monitor socio-economic differences in health indicators [93]. Given the direct connection, this could provide a good basis for identifying areas of epidemiological onset, in order to dispense directed and appropriate interventions.

In fact, the practice has shown that imposing regulations regardless of the epidemiological knowledge of the onset of the disease has very limited effects.

## 6. Facts and Patterns of the Onset of the Epidemic: the Relevance of Environmental Factors

New habits then [94] can lead to obesity in certain environments, and epidemiological studies [95] reveal that obesity occurs and develops systematically here and there, but mostly in low-income urban and suburban areas [96]. As a consequence, the recognition of obesogenic environments, the importance of which have been acknowledged for some time [97], can be verified through defined parameters. This means that we can avoid dispersed intervention in order to promote specific actions for the change of microenvironments with appropriate control and surveillance [98].

Getting people to change is not easy [99] and the role of public action remains that of modeling interventions in relation to the epidemiological evidence, in conformity with the aspects of health character and the administrative and accounting procedures of the territory, respecting the different needs of each specific area. The correct use of social accounting in the Local Health Units could play a very important role in the interception and monitoring of disadvantaged areas, thus enhancing the sustainability that these instruments can achieve for stakeholders.

Therefore, it is necessary to change the choices that affect the lifestyle of citizens in an effective manner. Some positive results to this aim seem to have been obtained through the specific public policies of *social marketing* [100] and *nudges* [101], despite the heated debates that the latter caused over time [102,103]. These policies have helpfully directed the dietary behavior of consumers according to various aims [104]. They have achieved this through a correction of certain characteristics of the environments in which citizens live and by concentrating on the reasons behind their unhealthy dietary choices. Even though some recent analyses demonstrate that nudge techniques can be an effective public health strategy for combatting obesity [105], these instruments can certainly not be considered a cure-all, but should be included in a framework of directed measures, in consideration of the multidisciplinary nature of the problem [106]: they can help to put in place a systematic action aimed at getting to know the benefits of the MD as well.

The nudge action can be structured in different degrees of effectiveness, in accordance with the aims to be pursued: providing information, communicating and informing citizens; enabling people to make good choices; educating them; leading them to the right choice through changing the default option; making the default option the healthier choice; using incentives to guide people to opt for the healthiest choice; using disincentives to discourage them from making harmful decisions; regulating, containing or definitively eliminating the prospects of choice. It can be applied in limited areas, concurrently with other measures aimed at improving the environment, contributing to positively affect the quality of citizens’ choices. Since it is a question of modifying environments, the effort must be multilevel and multi-competency and cannot be limited to isolated actions. In view of all of this, it is now clear that a single measure to tackle obesity will have a largely insignificant result, if it is implemented without an orderly vision of the epidemiological characteristics of the disease that it is put in place to control. 

The public law research on this topic is well advanced, and since the Nineties, some European governments, aware of the effects of the spread of obesity, have not hesitated to implement new proposals, like excises on unhealthy foods and traffic light food labeling. Of course, even if the law is considered to be a decisive tool for addressing childhood obesity and NCD in general [107], it seems appropriate to verify the effectiveness of some of these measures in light of the studies that have dealt with them. 

This paper therefore is proposed to proceed to a general examination of the currently known policies to tackling obesity, trying to highlight the criticalities that each regulatory intervention entails, as well as the potentially achievable advantages. 

The most widespread public policies adopted to reduce childhood obesity are schematically indicated in Figure 1.

## 7. Fiscal Strategies for A Healthier Lifestyle: Do They Work?

Excise taxes on junk food often generate strong debates when they are proposed in order to modify the nutritional choices of citizens. However, we cannot disregard a very brief examination of the effects of their application in a thorough assessment of what may be the most effective measures to spread the adoption of the MD as much as possible.

Very sugary drinks have no place in the MD and are unanimously considered to be harmful foods, but it is very difficult to eliminate them from the diets of adults and children. To reduce their consumption, governments have imposed excise duties on these products since 1920 [108], principally with the aim of generating revenue [109], but nowadays this sort of taxation is considered more as a social instrument to stop the increasing obesity rate. Therefore, they are currently in place in a number of countries [110], while elsewhere [111] they have had a short lifespan and only marginal results. 

The real objective of an excise of this kind should be the adoption of healthy regimes and the consequent reduction of obesity, but despite all the heated discussions generated, it is still not clear if they really can help to reach this goal.

For example, there is empirical evidence that connects taxation with a considerable decrease in sales [112], and yet there is still no clear evidence that taxation can lead to a real improvement in a population’s health. What is more, even if we were to have high quality data on the population’s health, which currently we do not, it would be very difficult to establish whether the improvement was due to a single specific political measure [113].

The results of some studies investigating the extension of a value-added tax to certain categories of food products have shown that it is indeed possible to obtain small health benefits, but unexpected results are also possible; for example, they have shown that taxing saturated fats leads to an increase in salt intake and greater overall mortality [114].

On the other hand, a connection between fizzy drinks and a large number of diseases has long been established: these diseases include cardiac and cerebrovascular disease [115] and reduction of bone strength [116].

Not only it has been hypothesized that fizzy drinks can contribute to causing erectile dysfunction, but it is also been highlighted that monosaccharide fructose is the most harmful sugar component in terms of weight gain and metabolic disturbances, while high-fructose corn syrup is gradually replacing sucrose as the main sweetener in soft drinks and has been accused of being potentially responsible for the current high prevalence of obesity. There is also considerable evidence that fructose, rather than glucose, can be the more damaging sugar component in terms of cardiovascular risk [117].

However, it should be noted that no one would think of reducing the intake of fructose from fresh fruit.

Even if it is well known today that soft drinks have detrimental metabolic effects and that their consumption should be limited, they only represent a very small part of the problem, given that the consumer frequently consumes them alongside other junk food, which is full of salt, sugar and trans-fats, almost all of which is industrially produced and processed. 

We are talking about food available at very low prices on the market, which we have come to associate with images of happiness and socializing due to the amount and invasive nature of obsessively repeated advertising, which for this reason, makes the food even more dangerous for minors. Another aspect of this food that is perhaps even more attractive than its price, is the fact that it is readily available almost everywhere, from large urban areas to smaller provincial towns [118], and it is ready prepared and can be eaten immediately. In today’s society, where time is an increasingly rare resource, this is an aspect that cannot be overlooked.

Therefore, we must ask ourselves whether the various sugar or soda taxes can truly represent a realistic solution that is both feasible and desirable in a context where the real purpose would be to encourage citizens to adopt voluntarily a healthy regime.

The answer is not immediately clear: some studies [119] have proposed an examination of the advantages and disadvantages of implementing a tax on junk food as an intervention to halt the increase of obesity in North America. The results have confirmed that although it is probable that modest excises produce substantial revenue, it is however improbable that these excices have any real influence on the rates of obesity. It is more likely that higher excises on soft drinks would have a direct impact on the weight of populations at risk, but it is less likely that they would be politically acceptable or sustainable.

As things stand, the efficiency of these taxes seems to be irrelevant because the consumption of soft drinks is a small part of overweight people’s diets and the drinks that can substitute for sugary drinks can often be even more calorific than the original soft drinks.

In fact, evidence shows that consumption often merely shifts from one unhealthy product to another: a reduction in soda consumption is shown to be completely offset by increases in the consumption of other high-calorie drinks [120], leaving the calorie count unchanged and meaning the action was undertaken in vain. Only replacing sugar-sweetened beverages with natural water is associated with a reduction in total calories and weight loss [121,122].

Moreover, if the aim of imposing taxation on soft sugary drinks lies in wanting to reduce the rate of obesity, then the increase in revenue should go towards financing projects and programs that would work towards this goal. This sort of taxation would then be worth considering more carefully.

Usually governments, for political convenience, don’t have the ability to direct the increase in revenue derived from these taxes to social aims and the consumers have little control over the final use of these funds. Indeed, the effectiveness of healthcare programs and subsidiaries allocated *ad hoc* are probably only a determining factor of the fiscal success in the fight against obesity if proposed in a well-devised, multidisciplinary framework.

What’s more, the analyses and contributions of some researchers [123] demonstrate that selective taxation distorts the market, reducing the consumer’s freedom of choice; and tends to have a regressive impact [83], impoverishing the less-advantaged social groups, confirming that the excises on sugary drinks usually only weigh upon the less advantaged classes.

Instead, these taxes need to be inserted into a wider plan, one that possibly includes incentives for healthy foods, gently pushing people to improve their choices. For example, some studies show that about a quarter of consumers claim to not eat enough fresh fruit, but that they would eat more fruit if it cost a little less; around a fifth of citizens would do the same with vegetables. 

The same studies show that, in reality, in some environments, the actual diet rarely coincides with the healthiest one, but this does not reduce the weight of public-healthism as an ethical and practical nutritional reference point, as it has the ability to generate a sense of mass guilt, even in areas where it cannot have an effect on healthy eating in practice [124]. Certainly, cost is an important factor when we talk about nutritional choices and the impact of the price of fruit and vegetables—the most important part of the MD—on the pocket of the consumer, should certainly be re-evaluated in the context of fiscal planning based on citizens’ lifestyle. This is also the line of the WHO, which recommends the use of financial incentives in a complete and coherent political context [125].

It is true that: “The cost of basic food commodities, such as corn and soy, is very low owing to economic strategies for food production including direct and indirect subsidies or tax advantages implemented as part of the farm bill. These crops are highly profitable because they are the raw ingredients of most processed foods and beverages. Corn and soy are also the primary feed for livestock; thus, the prices of beef and poultry are also very low by historic and international standards. By contrast, production of fruits and vegetables, which receives little governmental support, remains expensive” [95].

These circumstances make it difficult to achieve a healthy remodeling of behaviors and habits of individuals and families. A few cents more will not be enough to address the consumer in the right direction [126], as a buying choice is motivated by many other impulses. Indeed, recent scientific evidence shows that the cerebral neurochemistry, when in a state of sugar withdrawal, conditions behavior, pushing the consumer to satisfy their need in any way possible, in a worrying parallel to other addictions [127]. Moreover, high consumption levels of soft drinks containing sugar have long been associated with mental health problems among adolescents, even after adjustments have been made in research for possible confounders [128], this is why a small price difference is not relevant in effectively influencing behavior towards better choices, and even if it was a price problem, as already discussed above, the subject’s preference would simply turn towards other drinks, which are often no less calorific and just as harmful. 

Labeling systems seem to be equally insufficient in the positive conditioning of citizens’ choices. For a while different studies [129] have shown that the information directed at changing eating habits, communicated only through labeling, has provided limited results in the conditioning of citizens’ behavior and can even lead to excesses of consumption.

It is quite clear then, just how difficult it is to change nutritional choices on a large scale for wide sectors of the population with the only help of fiscal strategies. In reality, the areas that the legal instrument would need to touch upon are numerous and influenced by different causal factors. It is therefore necessary to swiftly examine the other aspects that political action must consider in order to halt and control the multifaceted and multilevel problem of obesity, particularly considering other actors that could participate in attempts to stop the spread of the epidemic.

## 8. The Role of the Food and Drink Industry

When buying processed foods, people are not able to clearly evaluate the food’s levels of saturated and trans fats, sugar and salt. Instead, they usually rely on brand reputation to determine their choice. In general, labels do not help with clear evaluation, as everyone reading them has different backgrounds. To be effective, they require the consumer to already have a high level of nutritional education and then also a willingness to devote the time to reading and reacting to labeling information while shopping.

It is said that the food industry uses too many ingredients that are deemed unhealthy because they contribute to the taste and the manufacturing process, and are relatively inexpensive: some Food and Beverage Corporations (hereinafter FBC) and manufacturers have often been held responsible for facilitating the explosion of obesity around the world [130,131].

In effect, governments’ relationships with industrial enterprises are an equally significant factor in the fight to prevent the spread of obesity, especially in terms of agreements for improving the ingredients in products and remodeling commercial communications.

In England the food industry has failed to hit its target of cutting sugar by 5% over the past year, with experts describing the results as hugely disappointing and suggesting the government may be forced to introduce a tax, as with sugary drinks [132]. A voluntary approach has been tried and tested with sugar and has been found to be lacking, as progress has been slow and too reliant on a handful of responsible manufacturers.

Obviously, it is very difficult to convince manufacturers to change their products to make them less harmful. It is made even more difficult in advanced countries by the presence of laws that protect the rights of profit margins, just as there are laws to protect the rights of the citizens’ health, with the same constitutional level of protection generating conflict.

It is well known that food and beverage manufacturers often ignore the rise in obesity rates, since their purpose—which they must report to their shareholders—remains that of tending to profit and seeking to increase sales as much as possible. Many unhealthy products represent a major source of profitability for some companies, so the spectre of government regulation is particularly threatening [133,134].

The conflict of interests is clear [135]. Many multinational corporations insist that they will help reduce obesity rates, but it must be noted that they do not promise to solve them, which leaves room for them to promote plans to improve nutrition, while simultaneously continuing to promote unhealthy products. It may be useful to remember the case of salt in Britain: in 2003 the food industry was invited to take voluntary measures to reduce salt in processed food products, including bread, hamburgers, pasta and biscuits. In 2018, Public Health England (PHE), published the first comprehensive report on the voluntary salt reductions that had been achieved by food producers and retailers [136]. Only slightly more than half of the goals had been reached. In fact, seven product categories did not meet any of the average targets set for them, these included ready-made meals, soups and meat alternatives, making them some of the largest contributors to salt intake with a huge variability in salt content. Many of the main contributors to salt are more frequently consumed by children, such as ham, fried potatoes, pizza, bread, breakfast cereals, beans, cakes, pastries, fruit pies and sauces. These foods are also more likely to be consumed by those in lower socio-economic groups. This report confirms that voluntary targets need comprehensive monitoring and guidance to be effective and, most crucially, that the food industry cannot, and must not, be made solely responsible for lowering our salt intakes. And this does not only apply to salt.

Depending on the choices of public policy, the government can opt for the targets to be made mandatory, with penalties if the targets are not met, or alternatively to choose to intensify communication and education for the most disadvantaged groups of consumers to make them aware of the risks of consumption.

In any case, it is easy to understand how the adaptation of a normative model for the current needs of reducing childhood obesity cannot be achieved whilst the obesity epidemic is growing so exponentially that it is unthinkable that it can be slowed only by preventing citizens from eating harmful foods by the use of regulation.

On the other hand, the same goal will not be achieved either by demanding producers to make their products healthier as quickly as possible. The spread of obesity is simply too rapid.

It is necessary, above all, to improve citizens’ daily nutritional options promoting knowledge. Encouraging appropriate choices that can lead to a greater adherence of the MD is just as important, as evidenced by the majority of the population, who already eat well due to simple cultural choices, avoiding eating sweets, fats and processed meats for every meal or drinking water instead of fizzy drinks. It is therefore necessary to ensure that citizens’ purchases respond to more rational and informed impulses, especially with regard to minors.

On this topic, it is widely assumed too that children are more susceptible than adults to persuasion by clever advertising and are more influenced by peer pressure. Junk food advertising is widely connected with poor nutrition, which has led to the policy response of banning junk food advertising for children [137].

To this end, limiting advertising for harmful foods could help, above all for the more impressionable bands of society such as minors, but this measure should also be accompanied by social marketing and appropriate education. Food marketing intentionally targets children who are too young to distinguish advertising from truth and induces them to eat high-calorie, low nutrient, but often highly profitable, junk food. Manufacturers and FBC are so successful in this marketing that business-as-usual cannot be allowed to continue without appropriate limits being imposed on them.

Such legislation, however, should be introduced in a wider framework of actions to combat childhood obesity [138]. This is an action that many European countries have already undertaken, given that it is now definitively established that the effects of these forms of advertising are extremely harmful [139]. A study drawn up by the WHO [140] indicates that existing policies and regulations are markedly insufficient when it comes to addressing the continuing challenges in this sensitive area. This is because they often tend to use too narrow definitions and criteria, meaning that standards are applied only to certain types of media, rather than to those with the largest audience of children. Furthermore, the complex challenges faced by cross-border marketing are hardly ever dealt with. The multiple causes of these situations are underpinned by both strong control and opposition from parts of the private sector, and the weakness of some self-regulation schemes.

## 9. The Nature of the Obstacles to the Adoption of the MD: Addressing Research

Generally, people prefer to eat everyday foods that correspond to predetermined patterns within their food culture, so that each food can satisfy an already known standard. The parameters of these standards are compatible with their lifestyle: not too expensive, with the nutrition they deem they need, prepared in the time they have and fitting to social norms with expected costs. Because trying a new food represents a risk of waste, simplifying seems to be the best option.

Attempts to steer consumers toward cheaper yet nutrient dense foods have encountered resistance and the main barriers to adoption are generally affordability, a lack of compatibility with taste preferences, and the belief that diet recommendations are often too time-consuming [141].

In this sense, the MD provides a socially acceptable framework for the inclusion of a set of fresh vegetables and dried fruit in a nutrient-rich everyday diet. Many of the products included in MD are themselves economical—except for some types of fish which can be replaced with cheaper ones—but consumers generally resist eating these less-familiar foods on a daily basis, particularly if they believe that doing so means abandoning their cultural heritage, or that those foods are associated with a different culture or a different social class.

As for this problem, there is evidence of a social gradient in diet quality: some studies demonstrate that generally, MD products are more likely to be consumed by groups of higher socio-economic status, while the consumption of refined grains, processed meats and added fats have been associated with a lower socio-economic status [142]. This fact can represent the real challenge, because the determinants of food choice are both complex and multifactorial, but food price is one of the most relevant determinants [143]. The lowest-cost diets, both energy rich and shelf stable, are also the least healthy, as composed by dry packaged and processed foods, likely to contain refined grains, added sugars, and added fats.

In many industrialized countries, there are socio-economic gradients in diet, so that those who are better off consume healthier diets than those less well-to-do. Almost everywhere the available evidence reveals that income affects food intake both directly and indirectly through the dispositions associated with particular social class locations [144]. Economic differences often generate cultural thresholds that constitute the most consistent problem: understanding how to overcome them and structuring adequate public policies is particularly important for resolving the problems faced by those that are most vulnerable to the choice of insecure food.

The multiple challenges involved in improving diet quality include helping people recognize the lower cost foods within their own cultural heritage, increasing the convenience and accessibility of lower cost, lower energy dense foods, and finally, doing so without sacrificing taste or enjoyment.

The MD represents a real intangible cultural heritage that flourished in a specific socio-cultural context, the dictates of which have become the biomedical model of a cultural representation [145], leading to a greater awareness of the importance of the social and cultural context in which a food model develops. This not only includes the choice of food in itself, but also corresponds particularly to the idea of conviviality, preparation and the pleasure of shared meals. The environment in which the MD was once consumed saw populations that were physically active enjoy a diet that basically rested on local plant foods. The challenges to promoting the Mediterranean Diet within and outside the Mediterranean region have been much researched and a significant point of evaluation has been how accurately the cultural ideals recognized by United Nations Educational, Scientific and Cultural Organization (UNESCO) [146] really represent the region and how culturally acceptable these ideals are to populations outside of the Mediterranean [147].

In particular, the act of eating together is seen to promote cultural identity and ensures social continuity. This special aspect of the MD represents an opportunity for social exchange, sharing time together, creating a special connection and promoting a space for community values and hospitality. A complete cultural process, expressed through the safeguarding of culinary techniques and the transmission of social values, is at the core of the life-style regime, along with the acknowledgment of pleasure as a fundamental part of a sustainable eating pattern. This point differentiates the MD from most dietary models focusing on the role of diet to meet biological needs rather than the role that foods can play as a vehicle for cultural processes and social interactions. Achieving the balance between pleasurable eating and health can obviously be more troublesome in the current environment, where media is a dominant presence and consumers are faced with an overabundance of calorific and processed food. In effect, economic growth and the globalization of food production are the reasons behind the desertion of the MD in countries where it once flourished [148].

It is clear that in the current obesogenic environment, reconstructing the same socio-environmental model is the most difficult challenge and the feeble voice of nutrition is almost lost amidst all the competing interests from the food industry and the perennially incumbent media. Living in cultural environments continually exposed to an excess of harmful foods makes it difficult opt out to healthy choices, while such choices remain almost non-existent or ineffective because they are dispersive, which means the clear identification of a correct nutritional modality is necessary. This is the reason why the structuring of an adequate and systematic health communication that takes these elements into account to be effective in contrasting childhood obesity and the sustainability of health systems now seems essential. Clearly indicating what is not suitable for long-term human nutrition and allowing consumers to explore their shared cultural food heritage through proper culinary education might represent a first step to restructuring, explicitly showing the benefits of joining the MD. Adapting traditional food preparation techniques to the modern world would also help to promote the adoption of this healthy lifestyle.

Gradually replacing harmful processed foods with high caloric density is a process that can be assimilated with proper communication, correct information and extensive education. If consumers are required to maintain a dietary pattern, they need to be clear on what that this pattern is.

Figure 2 shows a communication framework of actions aimed at enhancing the adoption of the MD.

As previously mentioned, there are clear links between low socio-economic status, poor health and obesity, and it is a priority that health messages be made accessible to all sectors of society. The precise balance between good nutrition, affordability, culture and acceptable social norms is an area that deserves further study [149] as it is directly connected to the problem of the consumer’s empowerment. Further research is also needed to gain clarity as to what pleasurable eating means to different bands of consumers [150] and whether the practice of eating convivially is accessible and relevant to general populations. If research needs to go beyond examining the validity of the nutritional components of the MD, more than anything else it has to explore the legitimacy of its cultural ideals.

## 10. Individual Choices and Actions to be Implemented

In view of the above, it is clear that the magic recipe to reduce obesity has not yet been discovered. Diet, exercise and behavior modifications will remain the cornerstones of obesity treatment for the foreseeable future, with a set of well structured public measures needed to increase their diffusion, acting in multi-competence, in a system of different levels of intervention.

In this context, where prevention remains the most obvious key, obesity continues to impose an economic burden on both public and private payers and the high public-sector spending for obesity remains a major cause for concern in terms of sustainability. Countering the continuing spread of the Western diet could be considered a primary goal of prevention, in favor of greater adherence to the MD.

Figure 3 shows the consequencences of a massive adherence to the principles of the Western Diet: the consequences of less adherence to MD and the connection between rising rates of obesity and undeniable rising medical spending. Without a strong and sustained reduction in obesity prevalence through more adherence to MD, major costs will be imposed on the healthcare system for the foreseeable future.

However, it must be kept in mind that is always up to the individual to formulate and continuously redefine the level of health and well-being that they intend to achieve. They must have the will to incorporate knowledge about what is healthy, together with the fulfillment of the consequent choices and actions to be implemented.

While on the one hand society tends to lack stable and shared orientations, it certainly opens the way to individual empowerment [151], which in turn leads to an awareness of the role of citizens in the social context: individual choices, after all, have consequences at both community and corporate level. 

Even though people often modify their behavior in unpredictable ways, they can still be helped to improve their basic standards: obesity-inducing lifestyles are just an adaptation to external factors, and if our aim is to decrease obesity or improve diet quality through external measures using regulation and a selected set of practices, external conditions must be properly changed, starting with the pricing of healthy foods, the cost of sport and recreational activities, improving microenvironments and urban environments [152], and the promotion of a heuristic approach to make it easier for citizens learning about healthy eating.

Precisely in consideration of these topics, preforming *ad hoc* forms of architecture through the modification of micro-environments in order to *nudge* subjects to make advantageous choices seems to be a step forward for reducing the problem [153,154].

Cities often concentrate risks and hazards that can promote the development of obesity, diabetes and their comorbidities. In order to tackle the challenge, policy makers and urban planners must understand how their communities and families live and work in order to develop the best interventions and start to engender a shift in attitude and culture in how they look at their own health. Education and culture play a decisive role here: taste has long been recognized as a question of education, learning, and therefore, of belonging [155]. The concept of culture itself [156] has recently acquired very significant meanings for the medical and scientific fields that are full of consequences for future political choices in healthcare. In effect, health communication is becoming an important tool for determining citizens’ behavior, because food is not only a necessity, but also, and above all, it represents a language, a means of communication, and therefore a privileged point of observation for the study of human and societal culture [157].

In order to be effective, the public policy approach must therefore be at the same time environmental and individual, effectively balancing its interventions [158]. It is increasingly being recognised that effective responses must go beyond interventions that only focus on a specific individual, social or environmental level and instead embrace system-based multilevel intervention approaches that address both the individual and environment.

The specific effects on different population groups should be considered in the design, modulation and monitoring of interventions, while also requiring a thorough understanding of the behavior and the target audience [159,160]. In effect, interventions should be targeted and based on relevant audience characteristics.

The scheme proposed in Figure 4. Summarily contains the instruments of action, where the indicated disciplines are to be considered among the most effective and diffused practices and models for individual/behavioral improvement of lifestyles [161,162], with the aim to create engagement even when there is resistance to change [163]. Their further analysis does not concern the subject of the present discussion.

## 11. Conclusions

In this scenario, having shown that the Mediterranean Diet (MD) prevents many diseases, the dissemination of its principles must be placed at the center of health policies.

For improvement to be effective, externality must also be pervaded by adequate and coherent communication, because in a dietary and food context, imperfect and asymmetric information is very often synonymous with poor efficacy.

In reverse, if well-conceived, carefully structured, implemented, and well-sustained public health communication programs have the capacity to elicit change among citizens by raising awareness, increasing knowledge, shaping attitudes, and changing behaviors and lifestyle [164], effectively influencing health outcomes.

Even if the evidence obviously suggests that poor diets and lifestyle cannot be changed by information policy alone, improved information is nevertheless essential for a conscious choice.

This means that policy measures should include the allocation of funds to expressly encourage a greater adherence to the MD in childhood environments through appropriate information and education. In consideration of the substantial evidence that supports its effects in chronic disease prevention and management, the Mediterranean Model could be designed while considering the needs of the period we are experiencing, of the current food supply, which is obviously different from that of the Sixties, and designing specific practical strategies to further spread and implement, considering that nutritional habits are homogenizing everywhere. In effect, the greater issue is that the importance of having the majority of a population following a healthy diet like the MD is hugely underestimated. In a general prevention context, speaking abstractly about generic dietary patterns is not what’s needed. Instead, one model needs to be unequivocally promoted, that which has been identified as having proved to be the most effective amongst many others in terms of the prevention of obesity-related diseases.

The aim is to translate the MD traditional principles, identifying the key food-based components, in order to be able to adapt them also to populations with different nutritional traditions, or in multicultural settings, as already successfully experimented [165]. In other words, the MD Model should represent the translation of the key elements of the traditional MD to populations outside the Mediterranean Region in order to increase the likelihood of acceptability and sustainability.

Encouraging people to adopt MD habits would not only be beneficial from a public health perspective, but above all it would be a concrete measure of intervention in terms of economic sustainability.

## Figures and Tables

**Figure 1 nutrients-11-01306-f001:**
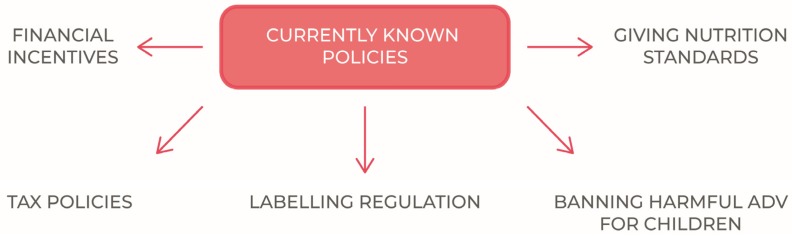
Currently known policies to tackle childhood obesity.

**Figure 2 nutrients-11-01306-f002:**
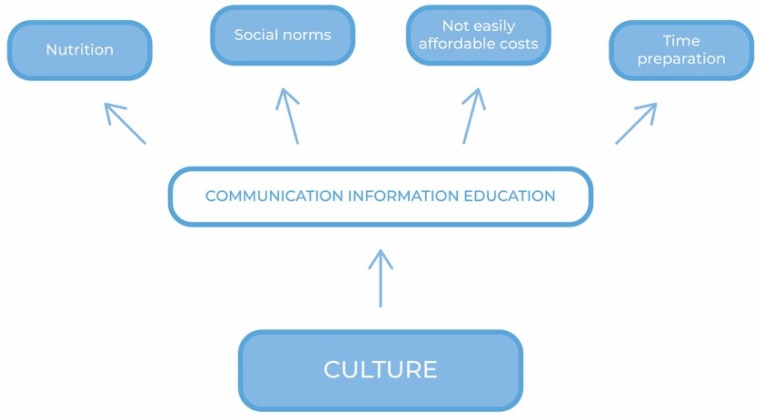
A communication framework of actions.

**Figure 3 nutrients-11-01306-f003:**
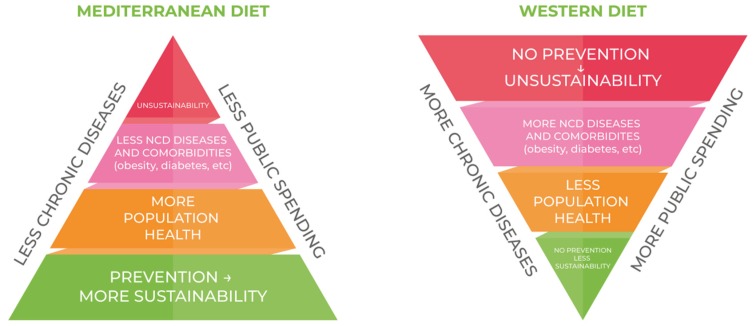
The consequences of the consumer’s daily choice.

**Figure 4 nutrients-11-01306-f004:**
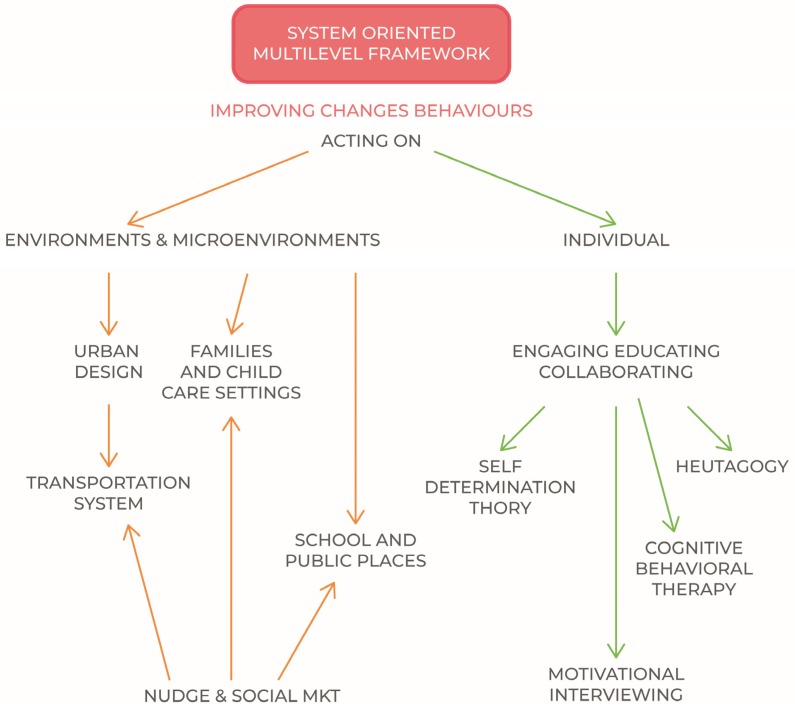
A system oriented multilevel framework.

**Table 1 nutrients-11-01306-t001:** Comparison of dietary foods between Mediterranean and Western Diets.

Foods	Mediterranean Diet	Western Diets
Vegetables	Every main meal (≥2 servings)	Rarely
Fruits	Every main meal (1-2 servings)	Rarely
Bread/pasta/rice/couscous/other cereals	Every main meal (1-2 servings, preferably whole grain)	Rarely whole grain cereals, often refined grains
Olive Oil	Every main meal (3-4 servings, expecially extra virgin)	Rarely olive oil, often replaced by margarine and butter
Nuts/seeds/olives	Every day (1-2 servings)	Occasionally
Dairy Foods	Every day in moderate portions (2 servings, preferably low fat)	Often high fat dairy foods
Herbs/spices/garlic/onions	Every day (less added salt)	Less often
Legumes	Weekly (≥2 servings)	Less often
Potatoes	Weekly (≤3 servings)	Less often
Eggs	Weekly (2-4 servings)	Less often
Fish/seafood	Weekly (≥2 servings)	Less often
White meat	Weekly (2 servings)	Less often
Red meat	Weekly (<2 servings)	Often
Processed meat	Weekly (≤1 serving)	Often
Sweets	Weekly (≤2 servings)	Often

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
