# Peer review of "Obesity and the Mediterranean Diet: A Review of Evidence of the Role and Sustainability of the Mediterranean Diet"

_nutrients, 2019, doi:10.3390/nu11061306_

Round 1
Reviewer 1 Report
In their manuscript, D’Innocenzo et al. present a review manuscript on the Mediterranean Diet and policy factors necessary to encourage better adoption of this diet. Overall, the manuscript tis well written and the topic is very important; however, some additions and minor edits could be made to encourage readability and novelty.
1) Please consider including a figure or table comparing the MD and Western diets for reference
2) Please consider including more data from non-Western countires…ie: Asia…and the difficulties with obesity and adoption of something like a MD in these places
3) Please consider including a table of factors impeding adoption of the MD and their advantages and limitations
4) Please consider including a table of currently known policies to encourage (or discourage) wider adoption of the MD and their feasibility and limitations
5) Please include a brief section on methodology used to generate this review including literature sources searched and search terms including which documents or papers with included or excluded and reasons for inclusion or exclusion
6) Lines 323, 333, 336, 383, 384, 420 : please fix references
7) Lines 489-493: While it is understandable that the authors are speaking abstractly, it would be advantageous to go further into how this “one model” would be designed and implemented
Author Response
REVIEWER 1
We thank the Reviewer for his/her suggestions all very relevant and consistent with the contents.
As suggested by the Reviewer we included a brief section on the Methodology used to generate our review pointig out the multidisciplinary of our approach (lines 31-51).
Following the indications, we included:
1) a table comparing the MD and Western diets for reference at line 220
2) more data from non-Western countiresand the difficulties with obesity and adoption of something like a MD in these places. On this poit, we added table n.5 (enclose in pdf and jpg, at line 714)
3) a paragraph about factors impeding adoption of the MD and their advantages and limitations (lines 566-643) and a table on the subject.
4) a table of currently known policies
5) Lines 323, 333, 336, 383, 384, 420 : fixed references
6) Lines 489-493: as suggested, we went further into how this “one model” would be designed and implemented, quoting best practices.
We thank again for the relevant insights.
Kind regards,
S, D’Innocenzo
C. Biagi
M.Lanari
Reviewer 2 Report
A nice overview of the causes and consequences of obesity with a narrative on the ideal preventative measures of which Med Diet plays a significant role.
I feel the title needs to be changed to better reflect what the paper discusses - obesity needs to be in the title.
Be careful not to overinflate statements e.g lines 106-109.
What was evidenced at the European Congress? Context needs to be provided here.
Not sure what ref 23 is referencing on line 67.
It is very wordy and would be a better read if condensed - in all sections. It can read like a thesis or text book and so needs to have a clear structure, connect the sections.
Author Response
REVIEWER 2
We thank the Reviewer for his/her suggestions all very relevant and consistent with the contents.
1) As suggested by the Reviewer we changed the title to better reflect what the paper discusses - obesity is now correctly in the title
2) We have changed several sentences not to overinflate statements in general.
3) We explained more clearly what had been evidenced more at the European Congress?
4) We made clear ref. 23 ref. on line 67.
5) All the text has been revised and condensed, made less wordy as possible, compatibly with respect of contents. Also the structure has been revised according to the precise and correct ndications.
We thank again for the relevant insights.
Kind regards
S, D’Innocenzo
C. Biagi
M.Lanari

Round 2
Reviewer 2 Report
The title does not reflect the content of the text. I'd suggest: Obesity and the Mediterranean Diet - a review of evidence of the role and sustainability. I don't feel there is enough of healthcare sustainability to warrant its presence in the title.
In the methods section - what is meant by 'a multidisciplinary approach'? The
The paragraph below does not make sense. What do you mean by multidisciplinary and multilevel? Please elaborate on this:
'The choice of a multidisciplinary and multilevel approach is due to the nature of the topic itself, which requires to be examined by different but complementary competences in order to be complete and adhering to the intended purposes. The need for conciseness imposes to conduct a revision activity which however must consider the multidisciplinary and the multilevel structure of the subject'
The aim of the paper needs to be stated up front early on in the introduction to bring the reader along with you.
